# Gamblers or Delegatees: Identifying Hidden Participant Roles in Crypto Casinos

## ABSTRACT

With the development of blockchain technology, crypto gambling has gained popularity due to its high level of anonymity. However, similar to traditional casinos, crypto casinos are controlled by a few internal *Delegatees*, making it impossible for them to achieve complete transparency and fairness. These delegatees are hidden among *gamblers* and are difficult to identify and distinguish in anonymous and large-scale blockchain transaction networks. This paper proposes an unsupervised dual-stage role identification method to adaptively identify key roles and hidden delegatees in label-sparse crypto casinos. Specifically, inspired by voting-style transaction patterns, we propose a novel voting influence metric for key node identification. This metric is based on one-dimensional structural entropy to capture global dissemination capability. Subsequently, we develop a multi-view graph neural network framework enhanced with two-dimensional global structural entropy minimization and self-supervised contrastive learning to improve the robustness and interpretability of hidden role partitioning. Experiments on real-world cases of the most mainstream blockchains—Ethereum, TRON, and Arbitrum—demonstrate that our proposed method effectively reveals distinct role compositions and collusion patterns, distinguishing between gamblers and delegatees. Our results achieve a higher match with identities confirmed by judicial authorities than existing methods, indicating the effectiveness and generalizability of our approach in enhancing security and regulation oversight.

## CCS CONCEPTS

• **Security and privacy** → **Economics of security and privacy**; • **Mathematics of computing** → **Graph algorithms**; • **Applied computing** → *Digital cash*.

## KEYWORDS

Crypto gambling, Anonymity, Security, Delegatees, Role identification, Graph Neural Networks, Structural Entropy

**ACM Reference Format:**
Anonymous Author(s). 2024. Gamblers or Delegatees: Identifying Hidden Participant Roles in Crypto Casinos. In *Proceedings of the ACM Web Conference 2025 (WWW'25), 2025, Sydney, Australia.* ACM, New York, NY, USA, 11 pages.

## 1 INTRODUCTION

The rapid growth of blockchain technology [61] has spawned a new form of online gambling: crypto casinos [9]. These crypto casinos offer various gambling games (e.g., poker, blackjack, roulette, slot machines) and allow players to bet using cryptocurrencies on multi-chain networks (e.g., Ethereum, EOS, TRON, Arbitrum)

*WWW'25, May 2025, Australia*

[4], providing a high degree of privacy and anonymity. In some crypto casinos, gambling funds or tokens strictly follow predefined rules set by smart contracts [63] to ensure perceived fairness and transparency, addressing issues like fictitious prize pools, opaque processes, high costs, and unpaid winnings.

However, the very anonymity afforded by blockchain technology renders crypto casinos less than absolutely fair and transparent, exposing them to security and compliance risks. A widely acknowledged issue in conventional casinos is that a few behind-the-scenes insiders or shills control and operate the entire casino [50], as shown in Figure 1. These roles, called *Delegatees*, are crucial in undertaking diverse tasks such as prize management, fund pooling, market promotion [56], and even illegal activities like money laundering and underground banking [51]. A similar but more challenging situation occurs in crypto casinos, where these delegatees exist hidden among gamblers and dispersed across large-scale and anonymous transaction networks, making them more difficult to identify. This circumstance leads to negative impacts, including misleading players, manipulating the market, and committing fraud [38]. Therefore, our purpose is to pinpoint these hidden delegatees, assisting players in reducing information asymmetry and herd behavior, and offering authorities insights for better regulation of crypto casinos.

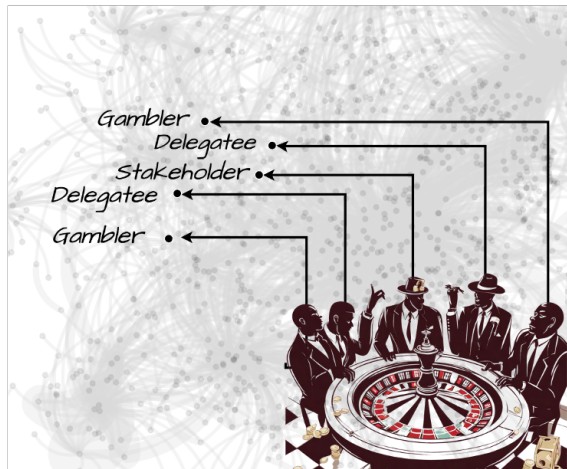

**Figure 1: Who are *Gamblers* or *Delegatees*?** Imagine gamblers gathered around a digital roulette table. Unbeknownst to them, a few behind-the-scenes *Delegatees,* cleverly hidden among gamblers, actually undertake diverse tasks and operate the entire crypto casino for *Stakeholders.*

Existing research on crypto-crime regulation and forensics offers valuable insights [25, 37, 55, 62]. However, these techniques primarily focus on address-level identification in extremely limited labeled scenarios. In reality, most anonymous addresses lack explicit labels, leading to an extreme imbalance with unlabeled addresses. Furthermore, existing address labels typically indicate the types of transaction scenarios entities are involved in, such as

exchanges, gambling, scams, DeFi, and mining, without delving further into the specific roles these entities play in their respective scenarios. Therefore, role-level identification allows for a more fine-grained understanding of entity functions and behaviors, especially in unlabeled casinos. Nevertheless, existing unsupervised role identification methods face three challenges: a) they are easily affected by the inherent randomness and noise of the graph, leading to a lack of robustness; b) most focus only on local network structures while ignoring global information; c) domain-specific knowledge has not been fully utilized, resulting in a lack of interpretability. Thus, it is imperative to adopt new approaches to identify the roles of different entities in crypto casinos.

This paper presents a novel approach named **C**rypto **C**asino **D**elegatee **M**iner (CCDM), to identify key and hidden participant roles in crypto casinos. CCDM organizes crypto gambling transactions using a multi-chain universal interaction graph model (§4.1) and then proposes an unsupervised dual-stage role identification algorithm for our label-sparse and noisy scenario (§4.2). Specifically, considering the voting-style transaction patterns and global structural information, we first develop a voting influence metric for key node identification using one-dimensional structural entropy (1D SE) (§4.2.1). Subsequently, for the remaining nodes, we design a multi-view GNN-based framework for adaptive hidden role identification (§4.2.2), in which local structural-view and temporal-view embeddings are initially obtained. To enhance the robustness of clustering (Challenge a), two-dimensional structural entropy (2D SE) minimization is incorporated into the loss function, synchronously serving as a measure of the global structure view (Challenge b). A self-supervised contrastive learning based on a feature similarity view is designed to improve the interoperability of clustering (Challenge c). Finally, a cross-role network motif extraction method is employed to analyze collusion patterns among roles (§4.3). Extensive experiments of real cases on Ethereum, TRON, and Arbitrum demonstrate that CCDM effectively uncovers distinct role compositions and ecosystems unique to each blockchain. Compared to existing methods, CCDM[1] results in a closer alignment with identities confirmed by judicial authorities and can be extended to other account-based blockchains. Our main contributions are as follows:

**(i)** We *first* presents a systematic investigation into the *security and transparency* of crypto casinos through the lens of role-level identification, providing a more fine-grained analysis of the ecosystem than address-level identification. **(ii)** We propose an *unsupervised dual-stage role identification* algorithm that combines key node identification based on 1D structural entropy and a multi-view GNN-based framework enhanced with 2D global structural entropy minimization and self-supervised contrastive learning to improve *robustness* and *interoperability*. **(iii)** Extensive experiments on real-world cases on Ethereum, TRON, and Arbitrum are conducted to uncover diverse role ecosystems and evaluate the *effectiveness* and *generalizability* of CCDM. **(iv)** Diverse types of collusion schemas among roles were extracted, such as *Arbitrage Triangle*, *Pooling*, *Listing*, *Sponsor*, *Prize Loop*, *Cross-Bridge*, *Staking* Schema, etc.

## 2 RELATED WORK

This section endeavors to investigate and provide a summary of recent relevant studies, which can be broadly categorized into three key research: crypto gambling, crypto-crime regulation and forensics, and role recognition research.

Existing research on crypto gambling provides the essential context for understanding this emerging form of gambling. Studies have explored the relationship between gambling and cryptocurrency trading [16], socioeconomic profiles of crypto gamblers [46], and user behavior patterns in decentralized gambling applications [39, 45]. Recently, emerging research has presented automatic tools [26, 53] to identify smart contracts and crypto addresses involved in gambling in large-scale blockchain networks. However, there remains a scarcity of research that delves into the identification and understanding of participant roles in crypto casinos.

Existing research on crypto-crime regulation and forensics provides us with key insights into how to secure cryptocurrency transactions and implement effective regulation of various types of crypto-related crimes, directly related to our research design. This work mainly focuses on account classification, abnormal address identification, and transaction tracing. To be specific, machine learning and deep learning methods are utilized to characterize behavior patterns of accounts and further achieve account de-anonymization [34, 37, 65]. Moreover, enhanced anomaly detection methods are employed to address more specialized threats, such as fraud [23, 24], phishing [11, 55, 57], mixing transactions [58], money laundering [36], and Ponzi [12]. Furthermore, novel transaction tracking tools have been proposed to study the flow of funds on blockchain networks [35, 59, 60]. However, these techniques mainly focus on address-level identification, which is limited to labeled scenarios. Most anonymous addresses lack explicit labels, resulting in a significant imbalance with unlabeled ones. Additionally, existing labels typically represent specific transaction scenarios (e.g., exchanges, gambling, scams, DeFi, mining), without examining the specific roles these entities play within those scenarios.

Role identification is directly related to our research objective and task. Intuitively, if two nodes have similar structures, they belong to the same role [43]. Unlike community detection, role identification focuses on the global structure and function of nodes, rather than just closely connected local groups. Current research is categorized into explicit and implicit role identification. Explicit role identification involves predefined, specific roles based on existing theories, experiences, or labels. These roles are typically classified using supervised or semi-supervised learning methods [20, 30, 64]. Implicit role identification does not require prior definitions and primarily uses methods like statistical analysis (e.g., block models [1], probabilistic models [13]), low-rank approximations (e.g., non-negative matrix factorization [22]), and unsupervised learning [41, 52]. Recently, both methods have been closely related to role-based graph embedding methods [19, 28, 29, 44]. Given the scarcity and extreme imbalance of labels in the anonymous blockchain scenario, we mainly focus on unsupervised methods. However, current unsupervised methods are easily affected by the inherent randomness and noise of the graph, leading to a lack of robustness. Most of them focus only on local network structures while ignoring global information. Domain-specific knowledge has not been fully utilized, resulting in a lack of interpretability.

---

[1]https://github.com/njublockchain/crypto-casino-delegatee-mining

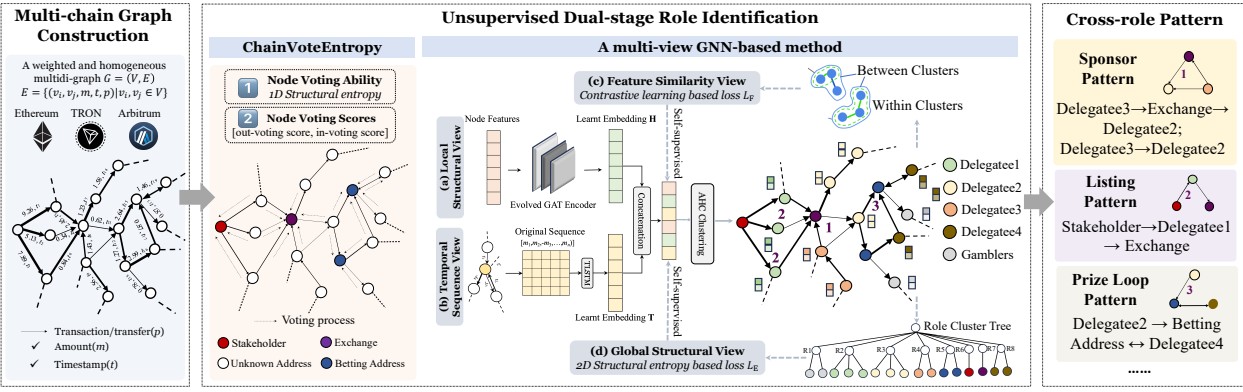

**Figure 2: A Framework of Crypto Casino Delegatee Mining (CCDM). First,** *Multi-chain Graph Construction* **creates a unified graph model. Next,** *Unsupervised Dual-stage Role Identification* **employs innovative ChainVoteEntropy and multi-view GNN methods to identify key and hidden roles respectively.** *Cross-role Pattern Analysis* **uncovers interaction patterns among roles.**

## 3 PRELIMINARIES

### 3.1 Problem Definition

Consider a weighted and directed graph $G = (V, E, \mathbf{X})$, where $V$, $E$ are the set of nodes and edges, and $\mathbf{X} \in \mathbb{R}^{|V| \times d}$ is the node feature matrix, with $d$ being the number of features for each node. The adjacency matrix $\mathbf{A} \in \{0, 1\}^{|V| \times |V|}$ represents the connections between nodes, where $\mathbf{A}_{ij} = 1$ if there is an edge between node $i$ and $j$, and $\mathbf{A}_{ij} = 0$ otherwise. Our task is twofold. First, the key role identification is to define an influence ranking function $f : V \rightarrow \mathbb{R}$, which assigns in-influence scores to each node. Second, leveraging the optimized node embeddings $\mathbf{Z}$, we perform clustering to assign the remaining node $v$ to a role cluster $C_v$. Notably, each node belongs to only one cluster and each cluster is linked to a unique role.

### 3.2 Structural Information Principles

In contrast to Shannon entropy, which measures uncertainty in probabilistic events, structural entropy extends this concept to the realm of network analysis, quantifying both the dissemination ability of nodes and the complexity of graph structure. This section mainly focuses on 1D and 2D structural entropy in a connected, undirected, and unweighted graph $G$.

**One-dimensional Structural Entropy (1D SE).** It captures the complexity of the entire network by focusing on the degree distribution and overall connectivity patterns, as defined below:

$$H^1(G) = -\sum_{v \in G} \frac{d_v}{2e} \log_2\left(\frac{d_v}{2e}\right) \quad (1)$$

where $d_v$ is the degree of $v$, and $e$ is the number of edges in $G$.

**Two-dimensional Structural Entropy (2D SE).** It delves into the modular structure of a network, considering both the community organization of nodes and the relationships between communities. Let $P = \{P_1, P_2, ..., P_L\}$ be a partition of the vertex set $V$ in $G$. The 2D SE of $G$ based on partition $P$ is defined as follows:

$$H^2(G) = -\sum_{l=1}^{L} \frac{V_{P_l}}{2e} \sum_{i=1}^{n_{P_l}} \frac{d_i^{(l)}}{V_{P_l}} \log_2 \frac{d_i^{(l)}}{V_{P_l}} - \sum_{l=1}^{L} \frac{g_l}{2e} \log_2 \frac{V_{P_l}}{2e} \quad (2)$$

where $L$ is the number of modules in the partition $P$. $n_{P_l}$ is the number of nodes in module $P_l$. $d_i^{(l)}$ is the degree of the $i$-th node

in $P_l$. $V_{P_l}$ is the volume of $P_l$, which is the sum of degrees of all nodes in $P_l$. $g_l$ is the edge count with exactly one endpoint in $P_l$ (i.e., edges that cross module boundaries).

## 4 A FRAMEWORK OF CCDM (FIGURE 2)

### 4.1 Multi-chain Graph Construction

Currently, crypto casinos primarily operate on account-based public blockchains like Ethereum and TRON, differing from Bitcoin UTXO model [10]. In the account model, addresses are either Externally Owned Accounts (EOAs) controlled by individuals or organizations, or Contract Accounts (CAs) managed by smart contracts [27]. This paper refers to all accounts by addresses. Interactions involve crypto asset transfers (e.g., ETH, TRX, tokens) between addresses, with each transaction having a single input and output address [31]. In the gambling ecosystem, project-specific tokens transfers are equally important. We track transaction flows and token transfers, and then construct complex interaction graphs.

**Definition 1. (Original Interaction Graph Model,** *og-IGM*): A weighted and heterogeneous multidi-graph $G = (V_{EOA}, V_{CA}, E_t, E_c)$, where $V_{EOA}$ and $V_{CA}$ represent the set of EOAs and CAs respectively. The EOAs set $E_t = \{(v_i, v_j, a, t) \mid v_i, v_j \in V_{EOA}\}$ represents the directed edge set constructed from transaction information. The CAs set $E_c = \{(v_i, v_j, f, t) \mid v_i, v_j \in V_{EOA} \cup V_{CA}\}$ represents the directed edge set constructed from token transfer information. The edge attributes $a$, $t$ and $f$ represent transaction amounts, timestamps, and token values respectively.

The *og-IGM* is a heterogeneous multi-graph that possesses dense connections as well as different types of information attached to nodes and edges. The heterogeneity of nodes and edges significantly increases the complexity of graph learning. Consequently, we further simplify the graph to a lightweight homogeneous graph.

**Definition 2. (Lightweight Interaction Graph Model,** *lw-IGM*): A weighted and homogeneous multidi-graph $G = (V, E)$, where $V$ represents the set of all addresses, and each node $v \in V$ has an extra attribute that indicates whether it is an EOA or CA. The edge set $E = \{(v_i, v_j, m, t, p) \mid v_i, v_j \in V\}$ represents interactions between addresses. Edge attributes $m$, $t$, and $p$ represent the interaction amount, timestamp, and whether the interaction is a transaction or token transfer, respectively.

## 4.2 Dual-stage Role Identification

In real-world networks, some nodes exhibit atypical features due to noise or anomalies. Hidden roles are often not immediately apparent, and directly analyzing all nodes increases the complexity of the task. Key nodes, which have greater influence and dissemination capabilities within the network, can be identified first to reduce redundant information. This provides more precise guidance and reference for subsequent hidden role identification, improving the overall efficiency and accuracy of the process.

*4.2.1 Key Node Identification.* Traditional centrality measures fail to fully capture the dynamic interaction pattern and global structure complexity of blockchain networks. We first draw inspiration from the voting-style behavior observed in token issuance and transfer, which can be seen as the inverse of node voting in networks. Second, as described in §3.2, 1D structural entropy not only considers the number of direct neighbors but also captures dissemination capability and coverage within the entire network. Combining these two aspects, we propose a novel indicator named **ChainVoteEntropy** (See Algorithm 1), designed for identifying key roles in directed and weighted blockchain transaction networks.

**Definition 3. (ChainVoteEntropy)**

**(i) Voting scores.** Different roles vary significantly in terms of directionality. For instance, *Exchanges* and *Service* often handle large volumes of both incoming and outgoing transactions, while *stakeholders* are typically involved in outgoing transactions. Thus, each node $v$ obtains its in-voting score $\tilde{N}_v^{\text{in}}$ and out-voting score $\tilde{N}_v^{\text{out}}$ separately from its input and output neighbors. The number of neighbors and the weighted sum of voting ability scores from neighbors of $v$ positively influence voting scores of $v$, defined as:

$$\tilde{N}_v^{\text{in}}(t) = \sqrt{\left( \sum_{(u,v)} b_u^{\text{out}}(t) \cdot m_{(u,v)} \right) \left| \tilde{E}_v^{\text{in}} \right|} \tag{3}$$

$$\tilde{N}_v^{\text{out}}(t) = \sqrt{\left( \sum_{(v,u')} b_{u'}^{\text{in}}(t) \cdot m_{(v,u')} \right) \left| \tilde{E}_v^{\text{out}} \right|} \tag{4}$$

where $b_v^{\text{in}}(t)$ and $b_v^{\text{out}}(t)$ represent the input and output voting ability of $v$ at the $t$-th iteration respectively. $m_{(v,u')}$ and $m_{(u,v)}$ represent transaction amounts from $v$ to $u'$ and from $u$ to $v$, respectively. $\left| \tilde{E}_v^{\text{in}} \right|$ and $\left| \tilde{E}_v^{\text{out}} \right|$ represent the number of in-neighbors and out-neighbors of $v$ respectively.

**(ii) Voting ability.** The voting ability of a node can be regarded as its capacity to disseminate information or influence other nodes. **1D SE** provides an effective metric for measuring voting ability. Specifically, nodes with higher 1D SE have their connections distributed across a larger and more diverse set of neighbors, resulting in more information dissemination paths and thus greater voting ability. Therefore, let each node $(\tilde{N}_v^{\text{in}}(t), \tilde{N}_v^{\text{out}}(t), b_v^{\text{in}}(t), b_v^{\text{out}}(t))$ is initialized as $(0, 0, H_v^{\text{in}}, H_v^{\text{out}})$, where $H_v^{\text{in}}$ and $H_v^{\text{out}}$ represent the 1D in- and out- structural entropies of $v$ respectively, as defined below.

$$b_v^{\text{in}}(0) = -\frac{wd_v^{\text{in}}}{V_G} \log_2 \frac{wd_v^{\text{in}}}{V_G}, b_v^{\text{out}}(0) = -\frac{wd_v^{\text{out}}}{V_G} \log_2 \frac{wd_v^{\text{out}}}{V_G} \tag{5}$$

where $wd_v^{\text{in}}$ and $wd_v^{\text{out}}$ represent the weighted (amounts) in-degree and out-degree of $v$ respectively. $V_G$ represents the sum of weights of all edges in $G$.

**(iii) Voting process.** In each iteration, selected key addresses that score higher between the highest $\tilde{N}_v^{\text{in}}$ and $\tilde{N}_v^{\text{out}}$ do not participate in subsequent elections. For in-neighbors of the selected node, their out-voting ability decreases by $\frac{1}{\langle k \rangle}$, where $\langle k \rangle$ is the average degree of $G$. For out-neighbors of the selected node, their in-voting ability decreases by $\frac{1}{\langle k \rangle}$. Other voting ability values remain unchanged. This process repeats until $p$ key nodes are selected.

*4.2.2 Hidden Role Identification.* **Manual Statistical Features.** These features are typically obtained through statistical analysis methods to reveal the characteristics of account states and historical transaction behaviors. Specifically, account state features include account balance, bounce, etc. Transaction intensity features encompass input/output transaction amounts, gas fees, etc. Transaction frequency features relate to the number of input and output transactions within a specific period, etc. Additionally, we derive aggregated features through operations such as summation, averaging, variance, Gini coefficient, etc. Detailed features are described in Table 5.

**Local Structural View (LSV).** In blockchain transaction networks, the number of neighbors, distribution structure, and transaction amounts have significantly different impacts on target nodes. Therefore, we propose an Enhanced Graph Attention Network (EGAT) that aggregates edge attributes with graph structures in latent representations. Specifically, the improved attention coefficient $\alpha_{ij}$ represents the importance of neighbor $j$ to $i$, defined as:

$$\alpha_{ij} = \frac{\exp\left( \text{ReLU} \left( \vec{a}^T \left[ W_h x_i \| W_h x_j \| W_e m_{ij} \right] \right) \right)}{\sum_{k \in \mathcal{N}(i)} \exp\left( \text{ReLU} \left( \vec{a}^T \left[ W_h x_i \| W_h x_k \| W_e m_{ik} \right] \right) \right)} \tag{6}$$

where $m_{ij}$ denotes transaction amounts between nodes $i$ and $j$, $\mathcal{N}(i)$ refers to the neighborhood of $i$, $W_h$ is a weight matrix used to transform node features, $W_e$ is a weight matrix for transforming edge features, $\vec{a}$ is a learnable weight vector, and $x_i, x_j$ represents the feature vectors of $i, j$ respectively. A reconstruction loss function $\mathcal{L}_G$ for EGAT is defined as:

$$\mathcal{L}_G = \| \hat{A} - A \|_F^2 \tag{7}$$

where $A$ is the original adjacency matrix, $\hat{A}$ is the reconstructed adjacency matrix, $\hat{A} = sigmoid(HH^\top)$, and $H$ is the learnt node embeddings from EGAT.

**Temporal Sequence View (TSV).** Transaction sequences occurring at different times represent distinct behavioral patterns. We utilize a Temporal Long Short-Term Memory (TLSTM) model to capture temporal features by arranging all transaction records for a given account in ascending chronological order. Each initial sequence is normalized to a fixed length $S$ by either padding with zeros or truncating. The plus or minus represents the direction of this transaction. The loss function $\mathcal{L}_T$ is defined as:

$$\mathcal{L}_T = \| \hat{T} - T \|_F^2 \tag{8}$$

where $T$ is the original sequence matrix, $\hat{T}$ is the reconstructed sequence matrix, and the total reconstruction loss function is:

$$\mathcal{L}_C = \mathcal{L}_G + \cdot \mathcal{L}_T \tag{9}$$

**Hidden Role Partitioning.** Gambling transaction networks consist of time-ordered transaction sequences, exhibiting hierarchical structures of users and fund flows. Then we input our concatenated embedding $Z$, which merges the *Local Structural View* and *Temporal Sequence View*, into Agglomerative Hierarchical Clustering (AHC) [6] for role partitioning. AHC is particularly suitable for datasets with hierarchical structures and uneven density distributions, characteristics common in gambling networks.

**Global Structural View (GSV).** However, due to the lack of predefined labels, we cannot guarantee the robustness of role results. On one hand, the graph noise and randomness are unfavorable for unsupervised learning. On the other hand, roles are collections of structurally similar entities, and currently, we only consider local neighborhood structures without considering the influence of global structures. To address these two issues, we employ 2D structural entropy minimization to enhance the robustness of role partitioning, while simultaneously capturing global structural information to optimize role partitioning. Given a role partitioning result $R = \{R_1, R_2, ..., R_L\}$ for G, we define:

$$\mathcal{L}_E = - \sum_{l=1}^{L} \frac{V_{R_l}}{V_G} \sum_{v \in R_l} \frac{wd_v^{in}}{V_{R_l}} \log_2 \frac{wd_v^{in}}{V_{R_l}} - \sum_{v \in V_{R_l}} \frac{g_l}{V_G} \log_2 \frac{V_{R_l}}{V_G} \quad (10)$$

where $L$ is the number of role clusters in $R$. $wd_v^{in}$ is the weighted (amounts) in-degree of node $v$. $V_{R_l}$ is the sum of the weighted in-degree of all nodes in $R_l$. $g_l$ is the number of edges that cross cluster boundaries in $R_l$. $V_G$ is the total weight of all edges in $G$.

**Feature Similarity View (FSV).** Another challenge of our role identification method is the absence of feedback about whether clusters are well explained. To address this challenge, we designed a self-supervised contrastive learning to achieve greater interpretability. Specifically, contrastive learning includes intra-cluster loss, which minimizes the sum of squared Euclidean distances within the same cluster to bring similar samples closer, and inter-cluster loss, which maximizes the sum of squared Euclidean distances between different clusters to push dissimilar samples farther apart.

$$\mathcal{L}_F = \frac{1}{|R_k|} \sum_{i,j \in R_k} \left\| Z_i - Z_j \right\|^2 - \frac{1}{|R_k||R_l|} \sum_{i \in R_k, j \in R_l} \left\| Z_i - Z_j \right\|^2 \quad (11)$$

where $R_k$ and $R_l$ represent two different clusters, $|R_k|$ and $|R_l|$ represent their number of nodes respectively. $Z_i$ and $Z_j$ are the concatenated embedding of node $i$ and $j$ respectively.

**Joint Training.** Finally, the model is trained by minimizing the total loss $\mathcal{L}$:

$$\mathcal{L} = \mathcal{L}_C + \alpha \cdot \mathcal{L}_E + (1 - \alpha) \cdot \mathcal{L}_F \quad (12)$$

where $\alpha$ is a coefficient controlling the balance in between.

*4.2.3 Time Complexity of CCDM.* Blockchain transaction networks are typically sparse graphs ($E \sim O(N)$) [5], where $E$ represents the number of edges and $N$ represents the number of nodes. *ChainVoteEntropy* primarily involves 1D structural entropy calculation and the voting process, with an overall time complexity approximating $O((N + E) + p \cdot (N + E)) \sim O(N)$, exhibiting linear time complexity. *Hidden Role Identification* primarily involves the training of LSV, TSV, GSV, and FSV views, with an overall time complexity approximating $O(E \cdot F + N \cdot S \cdot h + (N + E) + 1/c \cdot N^2) \sim O(N^2)$, where $F$ is the dimension of node features, $S$ is the input sequence length of

TLSTM, $h$ is the size of the TLSTM hidden layer, and $c$ is the number of clusters. Due to the distribution of nodes across different clusters, the actual complexity is lower than the theoretical upper bound $O(N^2)$. Compared to direct clustering methods, it has similar time complexity but achieves more fine-grained role partitioning.

## 4.3 Cross-role Pattern Analysis

Collusion or secret cooperation between different roles plays a crucial role in shaping the overall operation of crypto casinos. This section primarily reveals the intrinsic structure and relationships of transactions among roles and identifies the main collusion patterns that emerge from these interactions. We propose a cross-role pattern extraction algorithm (See Algorithm 2). Specifically, based on network motif methods, we extract motifs involving at least two different roles and calculate their occurrence frequency and z-scores [33], thereby obtaining frequent collusion cooperation patterns.

## 5 EXPERIMENTS AND CASE ANALYSIS

### 5.1 Experiment Setup

**Data Preparation.** This study focuses on three mainstream public blockchains: Ethereum, TRON, and Arbitrum to validate the effectiveness and generalizability of CCDM. Ethereum is currently the primary crypto platform. TRON, due to its low transaction fees, is the most popular blockchain for gambling activities to date. Emerging casinos are primarily concentrated on Layer 2 solutions such as Arbitrum, which offers faster transaction speeds and lower fees. These blockchains represent three mainstream operation mechanisms of crypto casinos to date.

Experimental datasets consist of native transaction, token transfer, and address-label datasets. Specifically, we use third-party software *Erigon* [17], *java-tron* [48], and *Nitro* [3] to synchronize the raw block data for Ethereum, TRON, and Arbitrum respectively. Then the open-source ETL tool *web3research* [40] is used to parse the raw data according to data structures (*Block*, *Transaction*, and *Token Transfer*). Labels of *crypto casinos* and *Exchanges* come from industry-trusted sources such as *etherscan* [18], *dappradar* [14], *tronscan* [49], and *arbiscan* [2], which are widely used in academic research (See §2). Other role labels are from manually verified labels by official law enforcement agencies in real-world criminal cases.

**Baselines.** CCDM is compared with existing unsupervised role identification: (1) Unsupervised graph learning (The generated representation from GNN-FiLM [7], SuperGAT [32], EGC [47], GATv2 [8], DirGNN [42] are each input into K-means [21] or DBSCAN [15] with the highest silhouette coefficient); (2) Low-rank approximations [22]; (3) Blockchain-domain state-of-the-art methods (Evolved PageRank with Local Community Detection (EPLCD) [59]).

**Evaluation Metrics.** To evaluate the effectiveness of CCDM, we use *Purity*, *Accuracy (ACC)*, and *Normalized Mutual Information (NMI)* [54]. *Purity* measures the extent to which clusters contain a single class. *ACC* assesses the match between predicted and true labels by finding the optimal one-to-one mapping. *NMI* quantifies the shared information between true and predicted clusters. See Appendix A.1 for details.

**Implementation.** In EGAT, *Output Embedding Size* is 32, *ReLU Negative Slope* is 0.2, *Attention Heads* are 4, and *Dropout Rate* is 0.5. In TLSTM, *Hidden Size* is 64, *Output Embedding Size* is 20, and *Sequence Length* is 15. *Layer* numbers in EGAT and TLSTM are treated

as variables. AHC uses *complete* linkage with *cosine* distance for high-dimensional sparse data. $\alpha$ are optimized by grid search. Each baseline GNN model has 2 layers with *ReLU*, *Output Embedding Size* of 32, and *Dropout Rate* of 0.5. All models use *Learning Rate* of 0.005. Training runs for 200 epochs with a batch size of 128, and the maximum number of role clusters is 10.

## 5.2 Case studies

**Ethereum Gambling.** We select multiple popular Ethereum casinos at that time and extract their complete set of gambling token transfers to verify the effectiveness of CCDM. Data statistics are shown in Table 1. First, we utilize the top-scoring nodes from *ChainVoteEntropy* to identify and verify key roles. Evidently, *Stakeholders*, engaged in token distribution, show higher output than input transaction volumes. *Exchanges* and *Service Providers* receive a relatively high volume of input and output transactions. *Fund Pools*, used for fund collection and management, exhibit higher input than output transaction volumes. Then we determined hidden delegatees and regular gamblers through Algorithm 4.2.2. Taking one of them, *Fairspin* (0xc2A8...580883), as an example, the predicted distribution of roles in the transaction network, the average values (*in-degree, out-degree, weighted in-degree, weighted out-degree, closeness centrality, betweenness centrality, PageRank*) of network features for each role, as well as 3- and 4-node network motifs with an absolute *z-score* greater than 2 [33], are shown in Fig 3.

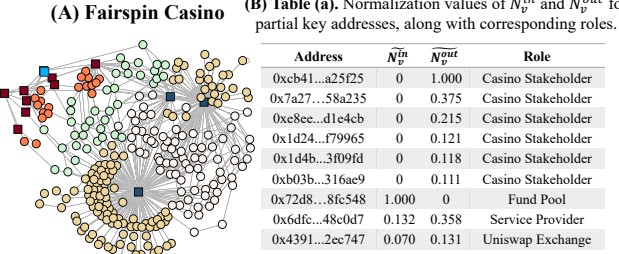

**(A) Fairspin Casino**

**(B) Table (a).** Normalization values of $\widetilde{N_v^{in}}$ and $\widetilde{N_v^{out}}$ for partial key addresses, along with corresponding roles.

| Address | $\tilde{N}_v^{in}$ | $\tilde{N}_v^{out}$ | Role |
|---|---|---|---|
| 0xcb41...a25f25 | 0 | 1.000 | Casino Stakeholder |
| 0x7a27...58a235 | 0 | 0.375 | Casino Stakeholder |
| 0xe8ee...d1e4cb | 0 | 0.215 | Casino Stakeholder |
| 0x1d24...f79965 | 0 | 0.121 | Casino Stakeholder |
| 0x1d4b...3f09fd | 0 | 0.118 | Casino Stakeholder |
| 0xb03b...316ae9 | 0 | 0.111 | Casino Stakeholder |
| 0x72d8...8fc548 | 1.000 | 0 | Fund Pool |
| 0x6dfc...48c0d7 | 0.132 | 0.358 | Service Provider |
| 0x4391...2ec747 | 0.070 | 0.131 | Uniswap Exchange |
| 0x74de...016631 | 0.010 | 0.006 | Airswap Exchange |

**(C) Table (b).** Network features of different roles, including *in-degree, out-degree, weighted in-degree, weighted out-degree, closeness centrality, betweenness centrality, and PageRank*

| Type | Role/Average Metrics | $d_{in}$ | $d_{out}$ | $wd_{in}$ | $wd_{out}$ | closeness | betweenness | PageRank |
|---|---|---|---|---|---|---|---|---|
| Key Roles | ■ Casino Stakeholders | 1.14 | **12.29** | 3.57E+08 | 7.14E+08 | 0.003 | 0.00036 | 0.0011 |
| | ■ Exchanges or Services | 42.50 | 52.63 | 8,654,029 | 5,167,423 | **0.220** | **0.09787** | **0.0450** |
| | ■ Fund Pools | **7.00** | 0.00 | 1.44E+09 | 0 | 0.204 | 0.00000 | 0.0057 |
| Hidden Delegatees | ○ Arbitrageurs | 2.66 | 2.40 | **518,361** | **518,074** | 0.195 | 0.00629 | 0.0030 |
| | ○ Listing Agents | 1.89 | 1.66 | 612,082 | 607,262 | 0.0932 | 0.00147 | 0.0018 |
| | ● Airdrop Promotors | 2.10 | 0.6 | 5.15E+07 | 1.82E+05 | 0.0052 | 0.00037 | 0.0012 |
| Gamblers | ○ Regular Gamblers | 1.89 | 1.16 | 491,229 | 421,968 | 0.1643 | 0.00133 | 0.0025 |

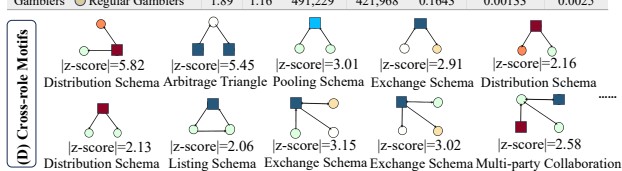

**(D) Cross-role Motifs**

| z-score =5.82 Distribution Schema | z-score =5.45 Arbitrage Triangle | z-score =3.01 Pooling Schema | z-score =2.91 Exchange Schema | z-score =2.16 Distribution Schema ...... |
|---|---|---|---|---|

| z-score =2.13 Distribution Schema | z-score =2.06 Listing Schema | z-score =3.15 Exchange Schema | z-score =3.02 Exchange Schema | z-score =2.58 Multi-party Collaboration |
|---|---|---|---|---|

**Figure 3: Role clusters of *Fairspin Casino*. CCDM not only identifies key roles (Stakeholders (3.3%), Exchanges or Services (1.1%), Fund Pools (0.3%)) but also uncovers hidden delegatees (e.g., Airdrop Promoters (6.9%), Listing Agents (13.6%), Arbitrageurs (30.5%)) and Regular Gamblers (44.3%). The connection patterns and activity modes among different roles exhibit significant differences in Table (b). Cross-role network motifs with an absolute z-score greater than 2 were extracted and identified as diverse collusion schemas.**

**Table 1: Data Statistics of Ethereum, TRON, Arbitrum Cases**

| | Node count | Edge count | Start_time | End_time |
|---|---|---|---|---|
| Ethereum | 467,615 | 2,486,531 | 09-21-2021 | 05-17-2023 |
| TRON | 1,287,360 | 4,354,765 | 09-09-2021 | 10-14-2023 |
| Arbitrum | 243,658 | 1,965,545 | 09-01-2021 | 08-29-2023 |

**Table 2: Normalization values of $\tilde{N}_v^{\mathbf{in}}$ and $\tilde{N}_v^{\mathbf{out}}$ for partial key addresses, along with corresponding roles. *Bonus payers*, serving as intermediaries, have more output transactions than input ones. *Exchanges* receive a relatively high volume of input and output transactions.**

| Address | $\tilde{N}_v^{in}$ | $\tilde{N}_v^{out}$ | Role |
|---|---|---|---|
| TVXn6N...tXXXXX | 0 | 0.863 | Bonus payers |
| TDDRAf...D88888 | 0 | 0.324 | Bonus payers |
| TQpLsV...eMMMMM | 0 | 0.089 | Bonus payers |
| TPRKHn...Zibb75 | 0.053 | 0.049 | OKEx Exchange |
| TQ6VZ7...KiSMeV | 0.023 | 0.025 | OKEx Exchange |
| TV866T...PoEfFJ | 0.016 | 0.021 | Binance Exchange |
| TEvgA6...9Z31J7 | 0.013 | 0.035 | Binance Exchange |
| TFfEJf...CR7qgh | 0.017 | 0.020 | Gate.io Exchange |
| TJHSgQ...z1rU7u | 0.005 | 0.006 | Gate.io Exchange |

**(a) Airdrop Promoters**: They may be individuals or teams involved in promoting and marketing gambling token airdrop campaigns. Their responsibility is to increase awareness of airdrop campaigns and attract more participants.
**(b) Listing Agents**: Listing agents serve as intermediaries between stakeholders and cryptocurrency exchanges. These agents are entrusted by *Stakeholders* to assist in the listing of their tokens on various crypto *Exchanges*.
**(c) Arbitrageurs**: Arbitrageurs are individuals or entities that engage in arbitrage trading, which involves buying and selling tokens or assets across different crypto *Exchanges* to profit from price discrepancies. The relatively balanced in-degree and out-degree of arbitrageurs demonstrate their role as bridges in fund transfers.
**(d) Regular Gamblers**: They are investors who have a conviction in the fundamental value and potential of particular tokens. They actively engage in the cryptocurrency market by purchasing gambling tokens on various *Exchanges*.

**TRON Gambling.** We focus on the most popular cross-border TRON casino "*Fānhuā Hash*". "*Fānhuā Hash*" operates by utilizing block hashes as the subject of speculation. Players use USDT-TRC20 to place bets. "*Fānhuā Hash*" offers a range of gambling rules, such as "Odd-Even Hash (TTwC4K...eM1Swm)", "Lucky Hash (TGDa2D...g9DDyU)", and "Bull Hash (TBaoBh...XdY6dD)", each of which corresponds to betting addresses. For simplicity of calculation, visualization, and analysis, we traversed multi-layer networks starting from one known betting address as a source node. Data statistics are shown in Table 1. After identifying key roles in Table 2, we discover diverse delegatees and regular gamblers beyond Ethereum casinos. Typical characteristics of suspected addresses for each role are shown in Table 3.

**Table 3: Statistical features of different roles on TRON casinos, including the number of input transactions($NIT$), the number of output transactions($NOT$), the total amount of input transactions($TAIT$)[Unit: USDT], the total amount of output transactions($TAOT$)[Unit: USDT], start time, and end time.**

| Suspected Addresses | Roles | $NIT$ | $NOT$ | $TAIT$ | $TAOT$ | Start_time | End_time |
|---|---|---|---|---|---|---|---|
| TVXn6N...tXXXXX | Bonus payers | 366 | 11,539 | 5,142,813.9 | 5,142,813.3 | 2022-07-27 | 2023-01-01 |
| TXTPLF...Y82zGH | Funding sponsors | 3,654 | 3,755 | 10,020,873.5 | 10,040,463.5 | 2022-05-09 | 2023-08-27 |
| TYe2Kt...DRouGL | Disguised gamblers | 6,947 | 9,494 | 14,299,163.9 | 14,298,197.1 | 2021-09-13 | 2022-09-13 |
| TGDa2D...g9DDyU | Betting addresses | 81 | 3 | 1,216.8 | 1,199.9 | 2022-08-05 | 2022-10-03 |
| TUuvLo...eCGf4Z | Real gamblers | 11 | 19 | 530.54 | 530.51 | 2022-09-04 | 2022-09-25 |

**(a) Fund Sponsors**: TRON casinos often send bonuses and transaction fees originated from *Fund Sponsors* for *Bonus Payers*. In this case, one of the suspected *Fund Sponsor* addresses (i.e. TXTPLF...Y82zGH) transferred 500 TRX to the bonus payer (i.e. TVXn6N...tXXXXX) on July 27, 2022. Evidently, the sponsor and bonus payer address belong to the same project team. The sponsor address has multiple deposit and withdrawal interactions with crypto *Exchanges* like Binance, OKEx, and Gate.io. They have a relatively balanced number of input and output transactions, as shown in Table 3.

**(b) Disguised Gamblers**: Among gamblers, some suspected addresses transfer funds to bonus payer addresses. Such gambler addresses are highly likely to be delegatees controlled by project team members. In this case, suspected disguised gambler addresses (i.e., TYe2Kt...DRouGL, TACJi9...anvGAY) exhibit the following unique characteristics, as shown in Table 3 and Figure 4: **(1)** Disguised gamblers send deposits to betting addresses. **(2)** Bonus payers send rewards to disguised gamblers. **(3)** Disguised gamblers return profits to bonus payers. **(4)** Compared to regular gamblers, they have an extremely large number of transactions and transaction amounts.

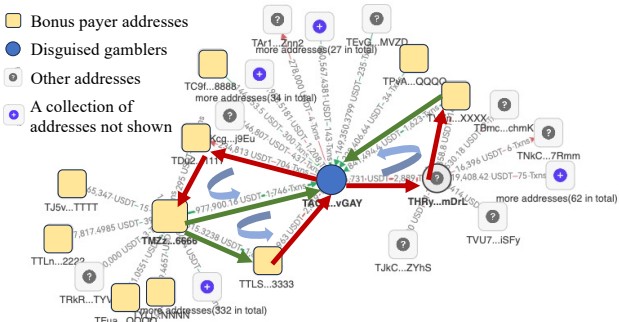

**Figure 4: Compared to *Regular Gamblers*, *Disguised Gamblers* return profits to bonus payers after bonus payers "pretend" to send rewards to them. Three *Prize Loops* exist in this chart.**

**(c) Regular Gamblers**: *Regular Gambler* (i.e., TUuvLo...eCGf4Z, TR5VMU...C86PUm) only transfers funds to known betting addresses and known bonus payer addresses transfer funds to them within a short period of time.

**Arbitrum Gambling.** *ZKasino* is a popular decentralized casino platform mainly operating on Arbitrum and Polygon. ZKasino eliminates the need for KYC procedures, user sign-ups, and traditional deposits or withdrawals. Instead, players retain full control of their funds in their own wallets, placing bets directly without any intermediary involvement. With more rapid transaction speeds and

lower fees, ZKasino is poised to revolutionize the online gambling industry. Each game on the platform corresponds to a smart contract address, including *Dice*, *Coin Flip*, *Plinko*, *Slots*, etc. For the sake of clarification, we select *Coin Flip Game* address(0xC4A4...E5279c) as the source node and extract its multi-layer transfers. Data statistics are shown in Table 1. We first identified *Uniswap*, *SushiSwap*, *Paraswap*, *Arbswap Exchanges* with both high $\tilde{N}_v^{in}$ and $\tilde{N}_v^{out}$ (i.e., 0x1B02...997506, 0x68B3...65FC45) and most likely gamblers with high $\tilde{N}_v^{out}$ (i.e., 0x2818...C0D2C7, 0xB28F...C9B282E) directly linked to the Coin Flip Game contract address. Notably, we discovered new delegatees in Arbitrum casinos beyond those found on TRON.
**(a) Cross-chain Bridges**: Cross-chain bridge is a technology framework or service that connects two or more different blockchain networks and allows assets to be transferred between them. Cross-chain Bridges (i.e., 0x80C6...99BCF8, 0xE4ED...00BCE8), involved in large volume transfers across networks, help gamblers transfer ETH back and forth between the Ethereum mainnet and Arbitrum.
**(b) Liquidity Providers**: *Regular Gamblers* or *Casino Stakeholders* tend to use diverse *Liquidity Providers* such as liquidity pools (i.e., 0x1054...ADE261, 0x652D...AC6D97), staking platforms (i.e., 0xEA8D...4C2176, 0x1F80...E58A28), and lending protocols (i.e., 0x2032...616B1F, 0xF4B1...0659E1) to exchange ETH for tokens that can be used for gambling. These *Liquidity Providers* handle frequent but smaller transactions. This is consistent with the staking-related activity and updates officially released by *Zkcasino*.

## 5.3 Model Performance, Ablation Studies, and Parameter Analysis

**Model Performance.** As stated in §5.1, role labels come from known *Exchanges* and some other roles verified by official law enforcement agencies in cases. As shown in Table 4, CCDM generally outperforms other methods across three blockchains regarding NMI, Purity, and Accuracy due to a dual-stage role recognition strategy and multi-view graph learning framework. Unsupervised Graph Learning (such as GATv2 and DirGNN) performs well, while EGC performs the worst in many cases, with different clustering algorithms yielding varying results. EPLCD shows moderate results, and RolX exhibits lower clustering performance.

**Ablation Studies.** To compare the performance with and without graph learning (0 on the horizontal axis), as well as the performance of LSV (1), TSV (2), GSV (3), and FSV (4) views within graph learning, we conducted an ablation study, as shown in Figure 5(a)(b)(c). Experimental results revealed that combining multiple views consistently leads to the best performance across Ethereum, TRON, and Arbitrum. GSV plays a crucial role in improving results, while FSV

**Table 4: Model evaluation(%) of our approach and baselines on Ethereum, TRON, and Arbitrum**

| Baselines | Ethereum | | | TRON | | | Arbitrum | | |
|---|---|---|---|---|---|---|---|---|---|
| | Purity | ACC | NMI | Purity | ACC | NMI | Purity | ACC | NMI |
| SuperGAT+KMeans | 75.00 | 68.75 | 67.06 | 70.59 | 58.82 | 64.75 | 69.23 | 53.85 | 53.20 |
| EGC+KMeans | 68.75 | 25.00 | 28.96 | 64.71 | 47.06 | 56.22 | 61.54 | 46.15 | 54.60 |
| GATv2+KMeans | 75.00 | 68.75 | 78.35 | 76.47 | 64.71 | 71.21 | 76.92 | 46.15 | 68.78 |
| GNN-FiLM+KMeans | 68.75 | 50.00 | 65.18 | 76.47 | 64.71 | 66.38 | 53.85 | 53.85 | 48.92 |
| DirGNN+KMeans | 62.50 | 43.75 | 64.85 | 70.59 | 58.82 | 62.48 | 76.92 | 61.54 | 58.94 |
| SuperGAT+DBSCAN | 68.75 | 62.50 | 65.52 | 70.59 | 64.71 | 67.55 | 61.54 | 61.54 | 54.11 |
| EGC+DBSCAN | 50.00 | 31.25 | 51.26 | 76.47 | 58.82 | 64.90 | 69.23 | 46.15 | 40.05 |
| GATv2+DBSCAN | 81.25 | 75.00 | 68.47 | 76.47 | 76.47 | 73.89 | 61.54 | 61.54 | 65.30 |
| GNN-FiLM+DBSCAN | 56.25 | 43.75 | 60.28 | 64.71 | 47.06 | 56.75 | 69.23 | 61.54 | 59.56 |
| DirGNN+DBSCAN | 75.00 | 56.25 | 67.06 | 70.59 | 58.82 | 67.90 | 76.92 | 69.23 | 73.03 |
| RolX | 68.75 | 62.50 | 42.98 | 64.71 | 52.94 | 55.92 | 61.54 | 46.15 | 42.20 |
| EPLCD | 68.75 | 50.00 | 59.26 | 76.47 | 58.82 | 64.30 | 69.23 | 69.23 | 64.65 |
| **Ours** | **81.25** | **81.25** | **79.66** | **82.35** | **82.35** | **78.34** | **76.92** | **76.92** | **71.52** |

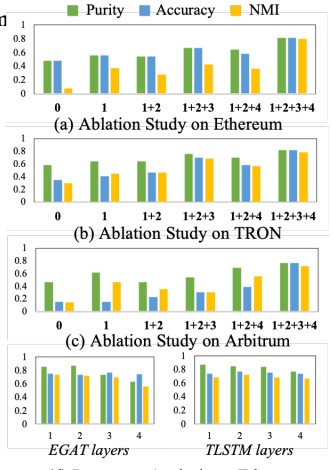

Figure 5: Ablation studies and parameter analysis

adds value but is less impactful than GSV. LSV provides a strong baseline but requires other views to reach optimal performance. Arbitrum, in particular, shows greater dependence on multiple views. **Parameter Analysis.** We explore the sensitivity of layers in EGAT and TLSTM and report experimental results on Ethereum in Figure 5(d). When the appropriate combination of parameters is chosen, it can overall improve the model performance, especially in terms of accuracy, which is of greater concern in real-world cases.

## 6 DISCUSSION

**Theoretic Implication.** Our study focuses on enhancing security and transparency in crypto gambling by uncovering key and hidden roles. Compared to address-level detection methods, our role-level discovery provides a more fine-grained analysis of entity functions without predefined labels. In terms of generalizability, CCDM can be extended to blockchain platforms employing the Ethereum Virtual Machine (EVM) model. In terms of scalability, CCDM is proven to exhibit near-linear time complexity in handling large blockchain ecosystems characterized by sparse graphs.

**Practical Implication.** This study elaborates on three distinct role compositions and operation mechanisms across the most mainstream blockchain ecosystems. *Ethereum casinos* closely resemble centralized casinos, using cryptocurrencies for transactions, thus the on-chain data consists of gambling token transfers. In this context, *Fund Pools*, as core account wallets, aggregate many deposits from users and transfer them to *Exchanges* for withdrawal. *Airdrop Promoters* are responsible for token distribution and community engagement. *Listing Agents* provide casino liquidity and exchange listing activities. *Arbitrageurs*, through frequent transfers among *Exchanges*, affect price fluctuations, market stability and player participation. **TRON casinos** conduct on-chain transactions for exchange, betting, and bonus payment using TRX and USDT. In this context, *Disguised Gamblers*, involved in fund transfers, may create false prosperity and mislead players. Additionally, if the design of gambling smart contracts is flawed, disguised gamblers might exploit vulnerabilities to influence game outcomes, ensuring casino profits while deceiving players. *Fund Sponsors* provide transaction gas fees or bonuses during casino operation. **Arbitrum casinos**, an Ethereum Layer 2 scaling solution, shares similar roles with TRON casinos but additionally requires *Cross-chain Bridges* for fund management and *Liquidity Providers* for casino operations.

For regulatory authorities, pinpointing key and hidden figures in crypto casinos enables effective regulation of illegal activities. For players, understanding hidden delegatees reduces information asymmetry, protects against manipulation, mitigates herd behaviors, and curbs deceptive practices. For casinos, we help optimize operations and risk management and enhance user trust.

**Future Work.** In the future, we aim to include data from emerging blockchain systems like Polygon and EOS for broader insights. At the same time, we will increase our focus on non-EVM platforms, such as converting the UTXO model to an account-based model using heuristic address clustering. Moreover, entities behind addresses may assume multiple roles, and these roles may evolve over time. We'll develop a hybrid role identification method based on structural entropy in overlapping community scenarios. Finally, we plan to explore richer transactional semantic information of smart contracts to provide a more comprehensive network analysis.

## 7 CONCLUSION

This paper proposes an unsupervised dual-stage role identification method to adaptively identify key roles and hidden delegatees in label-sparse crypto casinos, providing a more fine-grained analysis of the ecosystem than address-level identification. The method combines key node identification based on 1D SE and a multi-view GNN framework enhanced with 2D global SE minimization and self-supervised contrastive learning. Experiments on real cases of Ethereum, TRON, Arbitrum are conducted to show that CCDM not only quickly identifies key roles but also accurately uncovers diverse hidden delegatees and collusion patterns. CCDM achieves a higher match with identities confirmed by authorities than SOTA methods and is applicable to EVM-based blockchains. We help enhance regulation, player protection, and casino operations, promoting a more secure and transparent crypto gambling ecosystem.

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

# A APPENDICES

## A.1 Purity, Accuracy, and NMI in Clustering

*A.1.1 Purity.* Purity is an evaluation metric used to measure how well the clustering result matches the true class labels. It is computed by finding the majority class in each cluster and calculating the proportion of correctly assigned data points in the cluster. The formula for Purity is given by:

$$\text{Purity} = \frac{1}{N} \sum_{k=1}^{K} \max_j |C_k \cap L_j|$$

where $N$ is the total number of samples, $K$ is the number of clusters, $C_k$ represents the set of samples in cluster $k$, $L_j$ represents the set of samples in true class $j$, $|C_k \cap L_j|$ is the number of samples in cluster $C_k$ that belong to class $L_j$, and $\max_j |C_k \cap L_j|$ is the maximum number of samples from any single class $L_j$ in cluster $C_k$.

*A.1.2 Accuracy.* Accuracy is a common metric used in classification tasks, and when true labels are available, it can be used to evaluate clustering results. Accuracy measures the proportion of correctly classified instances. The formula for Accuracy is:

$$\text{Accuracy} = \frac{1}{N} \sum_{i=1}^{N} \mathbb{I}(y_i = \hat{y}_i)$$

where $N$ is the total number of samples, $y_i$ is the true label of the $i$-th sample, $\hat{y}_i$ is the predicted label (from clustering) of the $i$-th sample, and $\mathbb{I}(y_i = \hat{y}_i)$ is an indicator function that returns 1 if $y_i = \hat{y}_i$ and 0 otherwise.

*A.1.3 Normalized Mutual Information (NMI).* NMI is a metric that measures the mutual information between the predicted cluster assignments and the true labels, normalized by the average of the entropies of both the true labels and the cluster assignments. The formula for NMI is:

$$\text{NMI}(L, C) = \frac{2 \cdot I(L; C)}{H(L) + H(C)}$$

where $L$ is the set of true labels, and $C$ is the set of cluster assignments. $I(L; C)$ is the mutual information between the true labels and cluster assignments, defined as:

$$I(L; C) = \sum_{k=1}^{K} \sum_{j=1}^{J} P(C_k \cap L_j) \log \frac{P(C_k \cap L_j)}{P(C_k)P(L_j)}$$

where $P(C_k)$ is the probability of cluster $C_k$, $P(L_j)$ is the probability of class $L_j$, and $P(C_k \cap L_j)$ is the joint probability that a sample belongs to both cluster $C_k$ and class $L_j$.

$H(L)$ and $H(C)$ are the entropies of the true labels and cluster assignments, respectively:

$$H(L) = -\sum_{j=1}^{J} P(L_j) \log P(L_j)$$

$$H(C) = -\sum_{k=1}^{K} P(C_k) \log P(C_k)$$

NMI takes values between 0 and 1, where 1 indicates that the clustering perfectly matches the true labels.

## A.2 Manual Statistical Features

The detailed features are described below in Table 5.

## A.3 Key Node Identification

The detailed *ChainVoteEntropy* algorithm is described below in Algorithm 1.

## A.4 Cross-role Motif Analysis

The detailed cross-role network motif extraction algorithm is described below in Algorithm 2.

Received 13 Octobor 2024; revised 12 March 2025; accepted 5 June 2025

**Table 5: Manual Statistical Features include account state features, transaction intensity features, and transaction frequency features in Section 3.2.2. Account state features include account balance, bounce, etc. Transaction intensity features encompass input/output transaction amounts, gas fees, etc. Transaction frequency features relate to the number of input and output transactions within a specific period, the ratio between input and output transactions, etc. Additionally, aggregated features such as summation, averaging, variance, and Gini coefficient are derived.**

| Type | Symbol | Description |
|---|---|---|
| Account State | Balance | The amount of the cryptocurrency or token owned by the address |
| | Nounce | The total transaction amount of the address |
| Transaction Intensity | Total_amount | The total amount of all transactions of the address |
| | Total_in_amount | The total amount of input transactions of the address |
| | Total_out_amount | The total amount of output transactions of the address |
| | Avg_amount | The average amount of all transactions of the address |
| | Avg_in_amount | The average amount of input transactions of the address |
| | Avg_out_amount | The average amount of output transactions of the address |
| | Var_amount | The standard deviation amount of all transactions of the address |
| | Var_in_amount | The standard deviation amount of input transactions of the address |
| | Var_in_amount | The standard deviation amount of output transactions of the address |
| | Gini_amount | The Gini coefficient amount of all transactions of the address |
| | Gini_in_amount | The Gini coefficient amount of input transactions of the address |
| | Gini_out_amount | The Gini coefficient amount of output transactions of the address |
| | Gas | The transaction gas supplied by the sender of the address |
| | Gas_in | The gas supplied by the input transaction sender of the address |
| | Gas_out | The gas supplied by the output transaction sender of the address |
| Transaction Frequency | Num_all_tran | The number of all transactions of the address |
| | Num_in_tran | The number of input transactions of the address |
| | Num_out_tran | The number of output transactions of the address |
| | Fre_all_tran | The frequency of all transactions of the address |
| | Fre_in_tran | The frequency of input transactions of the address |
| | Fre_out_tran | The frequency of output transactions of the address |
| | R_in_out | The ratio of the number of input/output transactions of the address |

---

**Algorithm 2** Cross-Role Network Motif Extraction

**Input**: Graph $G = (\tilde{V}, \tilde{E})$, Role labels $R = \{r_1, r_2, ..., r_n\}$
**Parameter**: Node number of one motif $\tilde{S}$, Number of randomized networks $N$, Role number of one motif $\beta$, Threshold of z-score $\theta$
**Output**: Cross-role network motifs $\tilde{M}$

1:  Subgraphs $\tilde{g} \leftarrow \emptyset$
2:  **for** Each size $s \in \tilde{S}$ **do**
3:      $\tilde{g}_s \leftarrow$ FindSubgraphs$(G, s)$
4:  Motifs $\tilde{M} \leftarrow \emptyset$
5:  **for** Subgraph $g \in \tilde{g}$ **do**
6:      the set of roles: $\tilde{R}_g \leftarrow \emptyset$
7:      **for** Node $v \in g$ **do**
8:          **if** role$(v) \notin \tilde{R}_g$ **then**
9:              $\tilde{R}_g \leftarrow \tilde{R}_g \cup \{\text{role}(v)\}$
10:     **if** $|\tilde{R}_g| \geq \beta$ **then**
11:         $z \leftarrow$ z-score$(g, N)$
12:         **if** $z > \theta$ **then**
13:             $\tilde{M} \leftarrow \tilde{M} \cup \{g\}$
14: **return** Significant cross-role network motifs $\tilde{M}$

---

**Algorithm 1** ChainVoteEntropy Algorithm

**Input**: Graph $G = (\tilde{V}, \tilde{E})$, Edge weights $\tilde{W}$
**Parameter**: Output size $p$, Network Average Degree $\langle k \rangle$, Node voting abilities $ab_i, ab_j$
**Output**: Top-$p$ ranked nodes: $R$

1:  $\tilde{U}, \tilde{N}, R, \tilde{S} \leftarrow \emptyset$
2:  **for** $(v, u) \in \tilde{E}$ **do**
3:      $\tilde{U}_{(v,u)}, \tilde{U}_{(u,v)} \leftarrow a_i, a_j$
4:  $\tilde{S} \leftarrow \tilde{V}$
5:  **while** $|R| < p$ **do**
6:      **for** $v \in \tilde{S}$ **do**
7:          input score $\tilde{N}_v^{\text{in}} \leftarrow \sqrt{|\tilde{E}_v^{\text{in}}| \sum_{u \in (u,v)\tilde{E}_v^{\text{in}}} \tilde{U}_u \tilde{W}_{(u,v)}}$
8:          output score $\tilde{N}_v^{\text{out}} \leftarrow \sqrt{|\tilde{E}_v^{\text{out}}| \sum_{u \in (v,u)\tilde{E}_v^{\text{out}}} \tilde{U}_u \tilde{W}_{(v,u)}}$
9:      $v_{\max} \leftarrow \text{argmax}(\tilde{V}, \tilde{N})$
10:     pop$(\tilde{S}, v_{\max})$
11:     $R \leftarrow R \cup \{v_{\max}\}$
12:     **for** $(v, u) \in \tilde{E}_{v_{\max}}^{\text{all}}$ **do**
13:         Update voting abilities of the $v_{\max}$'s neighbors
            $\tilde{U}_{(v,u)} \leftarrow \max(\tilde{U}_{(v,u)} - \frac{1}{\langle k \rangle}, 0)$
14: **return** $R$

