# OpenReview forum: "Gamblers or Delegatees: Identifying Hidden Participant Roles in Crypto Casinos"
_ACM.org/TheWebConf/2025/Conference — WWW 2025 Oral_

### Official Review · Reviewer_1gnv · 2024-11-21

**Novelty:** 3
**Technical Quality:** 4

**Review:**

The paper introduces an innovative method, Crypto Casino DelegateeMiner (CCDM), which addresses the critical problem of identifying hidden participant roles in crypto casinos. By leveraging 1D and 2D structural entropy alongside a multi-view graph neural network (GNN) framework, the paper offers a fine-grained role-level analysis. This shift from address-level detection to role-level identification is a significant advancement in understanding the crypto-gambling ecosystem. The use of real-world data from Ethereum, TRON, and Arbitrum further enhances the relevance and applicability of the proposed approach. Overall, the paper is well-written and addresses an impactful problem. However, I have several major concerns about the approach, as outlined below.

Firstly, while the paper claims near-linear complexity for ChainVoteEntropy, the quadratic complexity of the role identification stages may become a bottleneck for large-scale datasets. It would be valuable to explore optimizations or alternative scalable methods to mitigate this issue. Additionally, I have reservations about the model's adaptability to the dynamic and rapidly fluctuating nature of crypto transaction networks, which are often influenced by external factors. The reliance on an extensive set of features raises questions about the model's robustness and adaptability in such scenarios. It is unclear how the model mitigates challenges related to feature dependency or dynamic environments, especially given the computational intensity of the feature engineering methods on large-scale networks. The model's structure, incorporating several substantial components such as EGAT, FSV, and TSV, results in a large and complex framework. This could potentially restrict its scalability and responsiveness to changing network dynamics. Moreover, the motivation for introducing the voting-based metric is not well described. The need for this new metric and how it addresses existing research gaps require clearer explanation, particularly in the introduction. Revising this section to highlight these aspects would greatly benefit the paper's narrative.

The ablation study, while included, could be emphasized further given the model’s multi-component nature. Specifically, the paper could explore the correlation among features to identify which features dominate and how they influence the outcomes. Currently, the reliance on existing features from previous research in fraud detection within crypto ecosystems makes the model’s novelty appear limited. For instance, features related to the multi-chain graph and EGAT have been explored in prior studies. It is crucial to clarify how this approach specifically addresses existing research challenges and gaps to establish its distinct contribution. Furthermore, the local view and global view have been studied extensively in the GNN world. So, what exactly novel here, it is not very well articulated. Lastly, the results section could be improved to better reflect the model’s size and complexity. Greater emphasis on critical insights from the results and how they validate the model’s robustness and efficiency would strengthen the overall impact of the paper. While the paper incorporates many valuable concepts and demonstrates potential, addressing the above concerns would enhance its clarity, robustness, and contribution to the field.

**Questions:**

1. How does the proposed CCDM framework address noise in the input graph, particularly in datasets with highly irregular transaction patterns? Given the rapid and fluctuating transaction behaviors in the crypto world, it is crucial to understand how the model ensures adaptability in such dynamic environments.

2. What measures are in place to ensure the robustness of the ChainVoteEntropy metric against adversarial manipulations, such as synthetic transactions designed to artificially inflate influence scores?

3. Could the authors elaborate on the role of the balance parameter (\(\alpha\)) in the joint loss function (\(L\))? Specifically, how does it manage the trade-off between global and local structure views? Additionally, have sensitivity analyses been performed to evaluate its impact on model performance?

4. How does CCDM scale with increasing data sizes, particularly in blockchains with high edge density, such as TRON? Understanding the scalability of the framework in such scenarios would provide valuable insights.

5. The motivation for introducing the voting-based metric is not sufficiently clear. Could the authors explain the specific need for this new metric and how it addresses gaps or limitations in existing approaches? Expanding on this in the motivation section would greatly enhance clarity.

**Reviewer Confidence:**

4: The reviewer is certain that the evaluation is correct and very familiar with the relevant literature

**Scope:**

4: The work is relevant to the Web and to the track, and is of broad interest to the community

---

### Official Review · Reviewer_zEUz · 2024-11-26

**Novelty:** 5
**Technical Quality:** 6

**Review:**

The paper presents an innovative approach, the Crypto Casino DelegateeMiner (CCDM), for identifying key roles such as players and hidden delegatees in blockchain-based casinos. By implementing a multi-phase, unsupervised learning strategy that integrates voting influence metrics, structural entropy, and graph neural networks (GNN), the study shows notable novelty and experimental success. The methodology was validated across mainstream blockchains—Ethereum, TRON, and Arbitrum—demonstrating both effectiveness and generalizability. Overall, this research holds significant theoretical and practical value, though certain areas could benefit from further refinement and expanded discussion.

**Strengths:**

- **Methodological Innovation**: The CCDM method uniquely combines voting influence, structural entropy, and GNN techniques. This approach is particularly novel in the context of role recognition in blockchain casinos, achieving breakthroughs in role differentiation and hidden role identification without labeled data.
- **Multi-perspective Analysis**: The paper introduces a multi-perspective GNN framework that includes local structure, temporal sequence, and global structure views. This design thoughtfully incorporates multiple feature dimensions to enhance classification accuracy.
- **Real-World Case Analysis**: Experiments were conducted using Ethereum, TRON, and Arbitrum, which are leading blockchain platforms. These tests confirmed the method's broad applicability and provided insights into the identification of delegatees and manipulation patterns in crypto casinos.

**Weaknesses and Suggestions**

- **Method Explanation**: The dual-stage unsupervised identification method includes several complex technical details, especially concerning the application of structural entropy and multi-perspective GNNs. Additional explanations and visualization could improve readability, particularly regarding the algorithm's rationale and design decisions.
- **Model Complexity and Module Contributions**: The model involves several modules (e.g., GNN, structural entropy, contrastive learning), adding to both complexity and training costs. The independent contributions of each module could be more explicitly validated. An ablation study could clarify each module's impact on model performance, facilitating a potentially more lightweight role identification method. In particular, further optimization of the global structure analysis within GNN could be beneficial.
- **Security and Privacy Concerns**: While the paper emphasizes transparency and security in identifying roles within blockchain casinos, the potential privacy impacts on users have not been fully explored. For instance, tracking hidden roles using structural entropy and voting influence metrics could inadvertently reveal user information. Privacy-enhancing mechanisms, such as differential privacy, might mitigate sensitivity to user data.
- **Context-Specific Blockchain Information**: The paper primarily focuses on graph structure features, leaving out blockchain-specific contextual information (e.g., smart contract features and transaction tags) that could aid in role identification. Attributes like transaction frequency and interaction patterns associated with smart contracts might reveal operational models in crypto casinos. A comprehensive role identification model, incorporating these features as node attributes, could distinguish role behavior patterns more effectively and improve identification accuracy.

**Questions:**

- In role classification with unlabeled addresses, has the possibility of introducing limited labels or adopting a semi-supervised learning strategy been considered to improve classification accuracy?
- The paper discusses improving transparency and security in blockchain casinos through role identification, but user privacy remains a key concern. Specifically, does the structural entropy and voting influence approach pose any risk to user privacy? Have privacy protection techniques like differential privacy been considered to reduce user data sensitivity?
- You mention that self-supervised contrastive learning enhances clustering interpretability. Are there specific interpretability metrics or case examples demonstrating how this method improves transparency and interpretability in practical applications?

**Reviewer Confidence:**

3: The reviewer is confident but not certain that the evaluation is correct

**Scope:**

4: The work is relevant to the Web and to the track, and is of broad interest to the community

---

### Official Review · Reviewer_aNjB · 2024-11-29

**Novelty:** 5
**Technical Quality:** 5

**Review:**

The paper introduces a novel unsupervised two-stage role identification method (CCDM), which combines structural entropy-based key node identification with a multi-view graph neural network framework. This enhances the capability to detect hidden roles within crypto casinos. The research provides valuable insights into improving the security and transparency of crypto casinos, aiding regulators in identifying and managing potential illicit activities. Experiments conducted on real-world data from major blockchains such as Ethereum, TRON, and Arbitrum validate the method's effectiveness. Results demonstrate its superiority over existing approaches in identifying roles and uncovering collusion patterns.

However, the use of intricate graph neural networks and multi-view learning results in high computational costs, which might be less accessible to resource-constrained researchers or smaller organisations. Also, although the paper claims near-linear time complexity, maintaining performance when dealing with extremely large-scale blockchain networks remains an open challenge.

**Questions:**

Does your model work effectively on other types of blockchains, such as Bitcoin's UTXO model? If not, how do you plan to adapt the model to accommodate these different environments?

Do you plan to explore alternative role identification methods in future research, particularly for application on non-EVM platforms like Polygon or EOS?

**Reviewer Confidence:**

3: The reviewer is confident but not certain that the evaluation is correct

**Scope:**

4: The work is relevant to the Web and to the track, and is of broad interest to the community

---

### Official Review · Reviewer_g73r · 2024-11-30

**Novelty:** 5
**Technical Quality:** 5

**Review:**

This paper proposes an approach to identify key and hidden participant roles in crypto casinos, called Crypto Casino Delegatee Miner (CCDM). CCDM combines one-dimensional SE-based key node identification with a multi-view GNN framework, while enhancing two-dimensional global SE minimization and self-supervised contrastive learning. Experimental results demonstrate that CCDM can identify key roles, hidden delegates, and collusion patterns in Ethereum, TRON, and Arbitrum. Compared to SOTA methods, CCDM achieves better results.



Pros:

1. The authors address the issues of security and transparency in encrypted casinos by proposing a more granular role-level identification method compared to traditional address-level identification. This approach facilitates a deeper analysis of the complex relationships and behavioral patterns within the encrypted casino ecosystem.

2. The authors propose an unsupervised two-stage role identification algorithm to address the issues of insufficient robustness, lack of global information, and poor explainability faced by existing unsupervised role identification methods.

3. The authors conducted experimental validation in real-world scenarios on the Ethereum, TRON, and Arbitrum platforms. The experimental results showcase the effectiveness of the proposed method.



Cons:

1. In the paper, the author selected the Agglomerative Hierarchical Clustering (AHC) algorithm to perform the "Hidden Role Partitioning" task and compared it against baseline algorithms K-means and DBSCAN. However, the paper lacks an in-depth analysis of the different clustering methods and does not adequately explain the specific advantages of choosing AHC. To strengthen the persuasiveness of the paper, it is recommended that the author provide a detailed discussion of the characteristics of each algorithm, especially the advantages of AHC over K-means and DBSCAN when handling "Hidden Role Partitioning".

2. The article lacks a detailed explanation of how the clustered addresses correspond to roles, especially in Section 5.2 case studies, where specific steps and validation methods are not clearly outlined. Additionally, the conclusions regarding multiple cross-role patterns do not detailed descriptions of the pattern recognition methods.

3. The text in Figure 4 is somewhat blurry, and the meanings of the red and green lines are not explained. Please enhance the clarity of the labels and provide a clear explanation for each colored line to improve understanding. These changes will significantly enhance the readability and interpretability of the figure.

**Questions:**

1. In the paper, the author selected the Agglomerative Hierarchical Clustering (AHC) algorithm to perform the "Hidden Role Partitioning" task and compared it against baseline algorithms K-means and DBSCAN. However, the paper lacks an in-depth analysis of the different clustering methods and does not adequately explain the specific advantages of choosing AHC. To strengthen the persuasiveness of the paper, it is recommended that the author provide a detailed discussion of the characteristics of each algorithm, especially the advantages of AHC over K-means and DBSCAN when handling "Hidden Role Partitioning".

2. The article lacks a detailed explanation of how the clustered addresses correspond to roles, especially in Section 5.2 case studies, where specific steps and validation methods are not clearly outlined. Additionally, the conclusions regarding multiple cross-role patterns do not detailed descriptions of the pattern recognition methods.

3. The text in Figure 4 is somewhat blurry, and the meanings of the red and green lines are not explained. Please enhance the clarity of the labels and provide a clear explanation for each colored line to improve understanding. These changes will significantly enhance the readability and interpretability of the figure.

**Reviewer Confidence:**

2: The reviewer is willing to defend the evaluation, but it is likely that the reviewer did not understand parts of the paper

**Scope:**

3: The work is somewhat relevant to the Web and to the track, and is of narrow interest to a sub-community

---

### Official Review · Reviewer_vqAs · 2024-12-01

**Novelty:** 5
**Technical Quality:** 5

**Review:**

The reviewer is against the development of gambling in any form.
The following comments are only made from the perspective of academic research.
In this manuscript, the authors propose a role identification method to identify key roles and hidden delegatees in crypto casino scenarios. The topic is interesting, and the proposed method is technically sound. Particularly, the extensive experiments on main mainstream blockchains validate the effectiveness and generality of the proposed method. Detailed comments are listed below.
Pros:
-Readability: The English usage of this manuscript is good. Also, the workflow and functions of distinct components are clearly described.
-Novelty: The proposed unsupervised dual-stage role identification method could adaptively identify key roles and hidden delegatees.

Cons:
-Organization: The overall organization is good. However, a clear threat model is much needed. Accordingly, the security properties should be formally defined. In addition, the security properties should be analyzed in a formally theoretical manner.
-A fraction of typos should be corrected. For example, on Page 3, Line 278 and Line 288, there is extra space.

The reviewer gives such a low rate in part because of ethical considerations.

**Questions:**

Except for the threat model, security definition, and security analysis issues mentioned above, the only question is:
Do you think there are ethical issues with the proposed techniques? Will it be used in an illegal business?

The reviewer gives such a low rate in part because of ethical considerations.

**Ethics Review Description:**

The topic is really interesting in practice. However, the reviewer is worried that the proposed techniques could be used to develop online casinos, which may further incur illegal issues, such as money laundering, human trafficking, etc.

**Ethics Review Flag:**

Yes

**Reviewer Confidence:**

4: The reviewer is certain that the evaluation is correct and very familiar with the relevant literature

**Scope:**

3: The work is somewhat relevant to the Web and to the track, and is of narrow interest to a sub-community